# Inverse Identification of Drucker–Prager Cap Model for Ti-6Al-4V Powder Compaction Considering the Shear Stress State

**Runfeng Li [1], Wei Liu [1,\*], Jiaqi Li [1] and Jili Liu [2]**

[1] School of Materials Science and Engineering, Wuhan University of Technology, Wuhan 430070, China; lrf980515@whut.edu.cn (R.L.); ljq2939@whut.edu.cn (J.L.)

[2] Hubei Key Laboratory of Theory and Application of Advanced Materials Mechanics, Wuhan University of Technology, Wuhan 430070, China; liujili@whut.edu.cn

\* Correspondence: weiliu@whut.edu.cn

**Abstract:** Numerical simulation is an important method to investigate powder-compacting processes. The Drucker–Prager cap constitutive model is often utilized in the numerical simulation of powder compaction. The model contains a number of parameters and it requires a series of mechanical experiments to determine the parameters. The inverse identification methods are time-saving alternatives, but most procedures use a flat punch during the powder-compacting process. It does not reflect the densification behavior under a shearing stress state. Here, an inverse identification approach for the Drucker–Prager cap model parameters is developed by using a hemispherical punch for the powder-compacting experiment. The error between the numerical and experimental displacement–load curves was minimized to identify the Drucker–Prager cap model of titanium alloy powder. The identified model was then verified by powder-compacting experiments with the flat punch. The displacement–load curves acquired by numerical simulation were compared to the displacement–load curves obtained through experiments. The two curves are found to be in good agreement. Meanwhile, the relative density distribution of the powders is similar to the experimental results.

**Keywords:** Drucker–Prager cap model; numerical simulation; inverse identification; shear stress; titanium alloy powder





## 1. Introduction

Titanium alloy has outstanding qualities, such as a low density, high specific strength, superior biocompatibility, corrosion and heat resistance, and so forth. It is widely utilized in airplanes, ships, and medical devices, among other things [1–3]. Meanwhile, it has earned the moniker "space metal", "bio-metal", and "ocean metal", among others [4,5]. Ti-6Al-4V is one of the most extensively used commercial titanium alloys and a typical representative of titanium alloy [6–8]. Powder metallurgy is a high-efficiency, material-saving material processing and preparation technology for titanium alloy [9,10]. In the field of powder metallurgy, component densification has a significant impact on the performance and service life of manufactured parts [11–13]. In the process of powder compacting, numerical simulation is one of the important methods to study powder densification. The powder constitutive model is the foundation and heart of numerical simulation [14,15].

In the numerical simulation of metal-powder compacting, there are two types of constitutive models. They are the discontinuous mechanics method and the continuum mechanics method. The discontinuous mechanics method is also known as the micromechanics method [16,17]. The powder is viewed as individual powder particles in this method, and the individual powder particles are modeled. It takes a long time to calculate. Large-scale modeling is not achievable in the real-computation procedure [18]. According to the continuum mechanics method, the powder can be approximated as a continuum deformer during the compacting process [19]. The elastic–plastic theory of materials was used

to investigate the deformation behavior of powders under compression circumstances [20]. This model contains many parameters, and the experimental approach is time-consuming. However, its simulation accuracy is high and it has a very promising application. The Drucker–Prager cap model (DPC) is a typical representative [21].

The Drucker–Prager cap model's shear failure surface and cap yield surfaces accurately describe the flow of the powder material due to shear and yielding due to plastic deformation during the compacting process [22]. It corresponds to the two densification mechanisms during metal powder compacting, namely, the rearrangement of powder particles and the plastic deformation of the powder itself. As a result, it is commonly used in the metal-powder-compacting process.

A series of studies have been conducted by relevant researchers for the numerical modeling of powder compacting and the Drucker–Prager cap constitutive model. Zhou [23] created a modified Drucker–Prager cap constitutive model based on pressed Ag57.6-Cu22.4-Sn10-In10 mixed-metal powders. It linked the model parameters with the relative density and validated the model's correctness using numerical simulations. Sinka et al. [24] used the Drucker–Prager cap constitutive model to investigate the effect of friction between the die and microcrystalline cellulose powder on the density of tablet compacts during the compacting of concave discs. The model was validated by comparing the relative density distributions of the simulated- and experimental-pressing outcomes. Melo et al. [25] employed the Drucker–Prager cap constitutive model to describe the alumina-powder-compaction process. To verify the correctness of the finite-element model, the experimental data of its two steps of closed-mold uniaxial pressing and isostatic pressing were compared with the results of the finite-element model. Hjalmar [26] performed tungsten-carbide-powder-compacting tests. Using the obtained machine pressures and finite-element calculations, the Drucker–Prager cap model parameters were inversely calculated. The method's feasibility was verified by employing density gradient measurements. Through uniaxial-powder-compaction testing with finite-element (FE) simulations in the ABAQUS, Atrian et al. [27] employed an artificial neural network (ANN) to derive the constants of the Drucker–Prager cap model for Al7075 powder. Morais et al. [28] used the method of digital-image-correlation techniques. Uniaxial and radial compression tests were used to determine the shear failure and elastic properties of the Drucker–Prager cap model for lead zirconate titanate powder (PZT). Using the Drucker–Prager cap model to analyze the creep of a pebble layer in a lava overburden, Yixiang Gan et al. [29] suggested a new approach for estimating the set of material parameters. Harona Diarra et al. [30] simulated the creep behavior of pharmaceutical powders using the Drucker–Prager cap model in ABAQUS. Also, the effect of viscoelastic behavior on the compaction mold was studied.

The inverse identification is convenient for determining the parameters of the Drucker–Prager cap model, but it should consider both the pressure and the shearing stress states. To solve this issue, we provide a new approach for identifying the parameters of the Drucker–Prager cap constitutive model in Ti-6Al-4V powder compacting. The error function is established using the displacement–load curve acquired from the compacting experiment with a hemispherical punch and the displacement–load curve obtained from the numerical simulation. The objective function is the error function, which is minimized. The parameters in the Drucker–Prager cap constitutive model were obtained by performing calculations using an optimization algorithm. It avoids complex experimental processes and allows the parameters of the constitutive model to be obtained quickly and easily. Simultaneously, the obtained relevant parameters were inserted into the finite-element model with the flat punch to obtain the displacement–load curve. To ensure the accuracy of the inverse identification parameters, it was compared with the displacement–load curves obtained from the experimental process.

## 2. Experiment and Simulation of Hemispherical Punch Compaction

### 2.1. Material and Experiment

The powder utilized in this paper is Ti-6Al-4V powder with a 400-mesh specification and a theoretical density of 4.5 g/cm$^3$. The powder was obtained from Beijing Xingrongyuan Company (Beijing, China). It was manufactured by the hydrogenation–dehydrogenation process. Its chemical composition is reported in Table 1.

**Table 1.** Chemical composition of Ti-6Al-4V powders.

| Ti | Al | V | Fe | C | N | H | O |
|---|---|---|---|---|---|---|---|
| 89.611% | 6% | 3.9% | 0.05% | 0.02% | 0.18% | 0.039% | 0.20% |

To obtain the displacement–load curve, compacting experiments on Ti-6Al-4V powder were performed using the universal material testing machine. The schematic diagram of the compacting experiment is given in Figure 1. In the compacting experiments, the molds were built of 60Si2Mn steel quenched at 850~870 °C, oil-cooled, and tempered at around 250 °C.

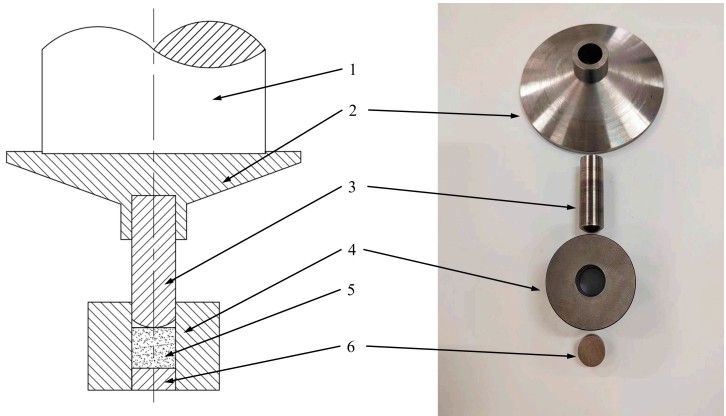

**Figure 1.** Tooling diagram of powder-compacting mold with a hemispherical punch; they should be listed as: 1—universal material testing machine; 2—amplifier; 3—hemispherical punch; 4—cavity die; 5—metal powder; 6—lower punch.

For the experiments, the powder was placed into a cavity die and compacted with a hemispherical punch. The green compact is shown in Figure 2. A universal material testing machine was used to obtain the displacement–load curves during the compacting process. Combined with the displacement–load curves obtained by numerical simulations, it prepares for the inverse identification of the parameters.

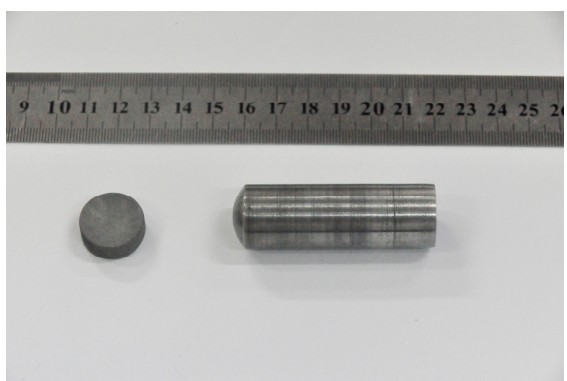

**Figure 2.** Diagram of powder samples compacted by a hemispherical punch.

### 2.2. Numerical Simulation

The compaction model was simplified in finite-element simulations for ease of calculation. The finite-element model is shown in Figure 3, and the ABAQUS axisymmetric is taken for the modeling. The hemispherical punch, lower punch, and cavity die were set as discrete rigid bodies, and the powder material was set as a deformable body. Next, meshing was performed, and the properties were set according to the experimental material.

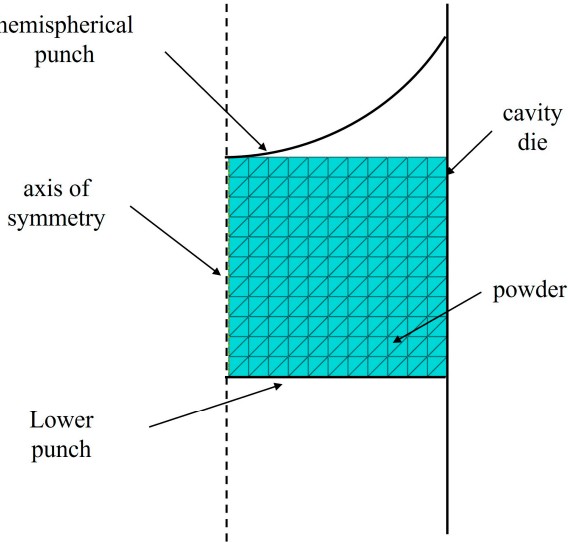

**Figure 3.** Schematic of finite-element model of the hemispherical punch.

The Drucker–Prager cap constitutive model from ABAQUS was used for parameter values. The modified Drucker–Prager cap constitutive model is isotropic. It is shown in Figure 4. The yield surface is made up of three parts: a linear shear failure surface $F_s$, which affects the material's deformation behavior under shear; a cap yield surface $F_c$, which represents the material's yield as a result of compression and also controls the material's unconfined shear expansion in shear; a transition surface $F_t$, to create smooth transitions to avoid numerical simulation instability [31,32].

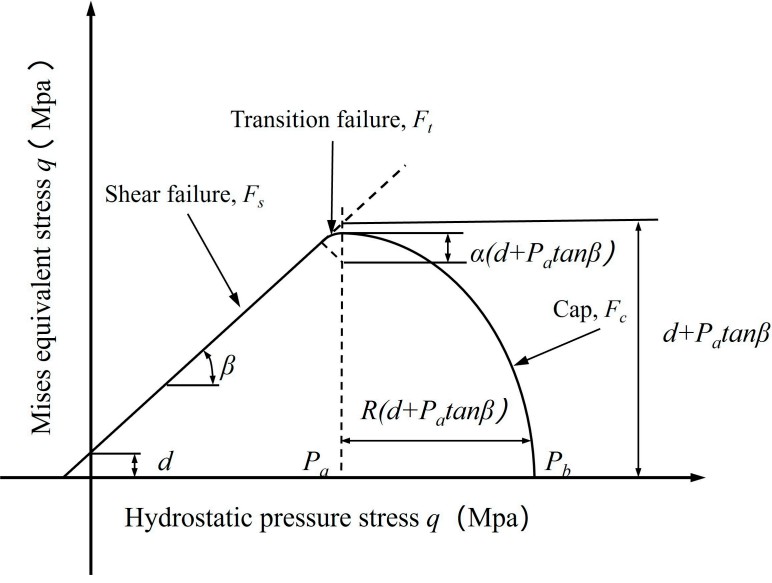

**Figure 4.** Schematic diagram of the Drucker–Prager cap constitutive model.

The linear shear failure surface $F_s$ is composed of two factors, material cohesion $d$ and angle of friction $\beta$, which characterize the shear deformation behavior of the powder body. It is expressed as:

$$F_S = q - p\tan\beta - d = 0 \tag{1}$$

where q is the Mises equivalent stress and p is the hydrostatic compressive stress.

The cap yield surface $Fc$ defines the material's yielding owing to compression, and also limits the material's unconfined shear expansion under shear. It consists of three parameters: cap eccentricity R, volumetric plastic strain Pa, and material-hardening yield stress Pb. The cap yield surface primarily reflects the nonlinear-hardening yield criterion of the material under powder compacting. It is expressed as follows:

$$F_c = \sqrt{(p - p_a)^2 + \left[\frac{Rq}{1 + \alpha - \alpha/cos\beta}\right]^2} - R(d + p_a \tan\beta) = 0 \tag{2}$$

where $R$ denotes the eccentricity distance, which controls the shape of the cap yield surface and has a value between 0.0001 and 1000; $\alpha$ is the transition coefficient, which is the transition surface's control parameter. The yield stress $P_b$ parameter is the value of the intersection of the cap yield surface with the transverse coordinate axis, which is used to determine the position of the cap yield surface. The volumetric plastic strain $P_a$ is then the value of the transverse coordinate of the intersection between the cap yield surface and the model transition surface, which indicates the material's plastic deformation behavior.

The transition surface $Ft$ smoothly connects the linear shear failure surface and the cap yield surface of the model. It allows the numerical analysis calculation process to successfully reach the convergence condition. The expression is as follows:

$$F_t = \sqrt{(p - p_a)^2 + \left[q - \left(1 - \frac{\alpha}{cos\beta}\right)(d + p_a \tan\beta)\right]^2} - \alpha(d + p_a \tan\beta) = 0 \tag{3}$$

where $\alpha$ generally takes the value range of $0.01 \leq \alpha \leq 0.05$. In this paper, it takes 0.02.

The Drucker–Prager cap model's shear failure surface parameters (material cohesion $d$ and angle of friction $\beta$), cap yield surface parameters (cap eccentricity $R$, volumetric plastic strain $P_a$, and material-hardening yield stress $P_b$), and powder material elasticity parameters (modulus of elasticity $E$ and Poisson's ratio $\nu$) can all be expressed as functions of the relative density ($\rho$). In the traditional approach, the relevant parameters are mainly obtained through experiments. A set of experimental data at various relative densities was obtained using uniaxial compression, radial compression, and uniaxial molding experiments. The functional equation between the parameters and the relative density was obtained by curve-fitting. Table 2 shows the functional equation for each of its parameters.

**Table 2.** Expression of Drucker–Prager cap parameters.

| Model Parameter | Abstract Expression |
|---|---|
| $d$ (material cohesion) | $d = d_1 e^{d2\rho}$ |
| $\beta$ (angle of friction) | $\beta = m_1 + m_2 e^{m3\rho}$ |
| $R$ (cap eccentricity) | $R = r_1 + e^{r2\rho}$ |
| $P_b$ (material-hardening yield stress) | $P_b = b_1 + b_2 e^{b3\rho}$ |
| $E$ (modulus of elasticity) | $E = e_1 + e_2 e^{e3\rho}$ |
| $\nu$ (Poisson's ratio) | $\nu = \nu_1 + \nu_2 e^{\nu3\rho}$ |

To improve the inverse identification of the parameters, the sensitivity of the model parameters should be investigated before performing the inverse identification. Because the upper punch in the traditional powder-pressing experiment is flat, the shear stress of the powder during the pressing process is very small. The inverse identification of the cohesion $d$ and friction angle $\beta$ is not very sensitive during traditional powder-compacting

experiments. As a result, the upper punch in the compacting experiment in this paper was designed as a hemispherical punch. The relevant parameters in the constitutive model are inversely identified based on displacement–load curves during the compacting process.

The model parameters' sensitivity analysis was carried out as follows. During the numerical simulation of powder compacting, the values of each parameter were adjusted to study the effect on the displacement–load curve, while other model parameters remained constant. As a result, the powder-compacting process must first be numerically simulated. The relevant parameters were set based on the compacting experiments, and numerical simulations were performed. The sensitivity of each parameter is shown in Figure 5. In Figure 5, the $d$, $E$, and $P_b$ represent the material cohesion, modulus of elasticity, and material-hardening yield stress, respectively. Their units are megapascal (MPa). The $\beta$ represents the angle of friction. It is the angular unit. The $R$ and $v$ represent the cap eccentricity and Poisson's ratio, respectively. They are numerical values without any physical unit.

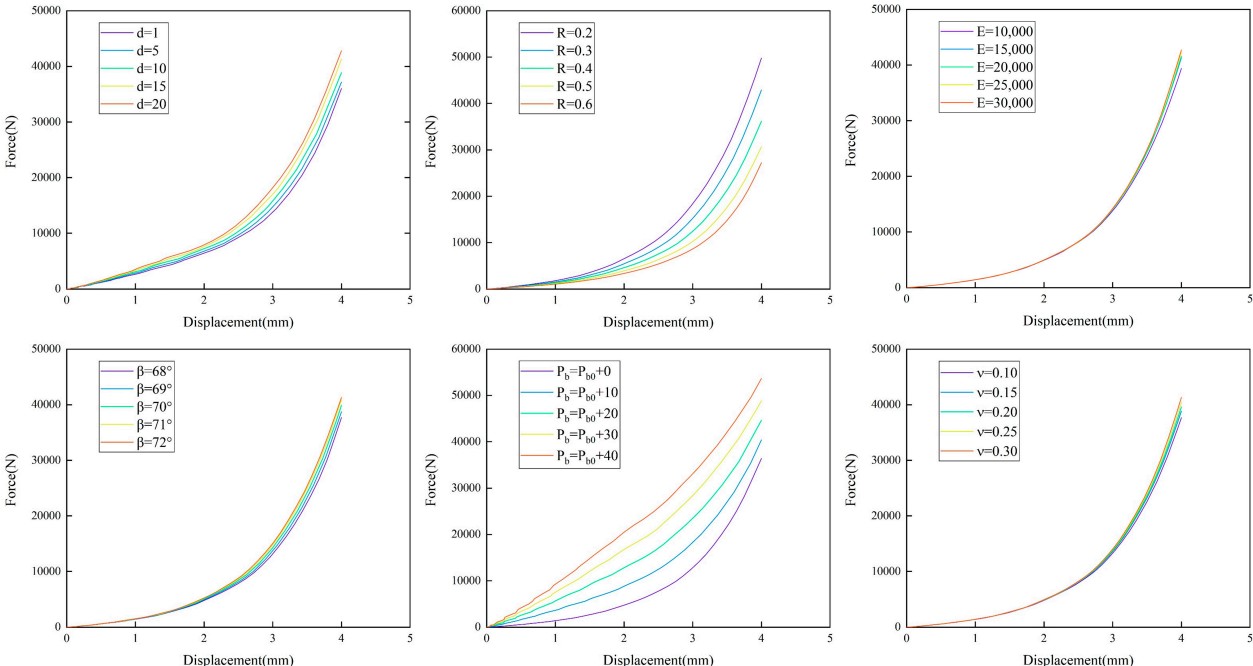

**Figure 5.** Sensitivity analysis of the model parameters.

As can be seen from the figure, every parameter has a certain sensitivity. The sensitivity of the shear failure surface parameters and the elasticity parameters is lower than that of the cap yield surface, but it is nevertheless present. Therefore, it is feasible to identify the parameters of the DPC constitutive model by the inverse identification of the displacement–load curve for the hemispherical punch.

## 3. Parameter Identification of the Drucker–Prager Cap Model

### 3.1. Inverse Identification

In obtaining the parameters of the Drucker–Prager cap constitutive model using conventional experimental methods, a series of powder destructive experiments are required. It is a costly, cumbersome, and time-consuming operating process. To simplify the process, a process of inverse identification of the parameters was established. The approach uses the model parameter expression coefficients as input variables. The error function is established using the displacement–load curve acquired from the compacting experiment with a hemispherical punch and the displacement–load curve obtained from the numerical simulation. Simultaneously, the error function is minimized. The shear failure surface parameters ($d$ and $\beta$), the cap yield surface parameters ($R$, $P_a$, and $P_b$), and the elastic parameters of the powder material (modulus of elasticity $E$ and Poisson's ratio $v$)

are the inverse identification model parameters. When paired with Table 1, the reverse identification technique input variables are (*d*1, *d*2, *m*1, *m*2, *m*3, *r*1, *r*2, *b*1, *b*2, *b*3, *e*1, *e*2, *e*3, *v*1, *v*2, *v*3). The displacement–load curve acquired from the compacting experiment and the displacement–load curve generated by numerical simulation form the goal function. It is used to determine the error function Fe, as stated in Equation (4):

$$F_e = \frac{\sqrt{\frac{1}{n}\sum_1^n \left(F_{exp} - F_{sim}\right)^2}}{\sqrt{\frac{1}{n}\sum_1^n \left(F_{exp}\right)^2}} \tag{4}$$

where $F_{exp}$ is the compression force from the compacting experiment, $F_{sim}$ is the compression force obtained from the numerical simulation, and n is the number of data points, which were collected using linear interpolation for comparison.

The flow chart of inverse identification is shown in Figure 6. Compared with the traditional parameter inverse identification method, the method proposed in this paper acquires parameters more comprehensively. At the same time, there is no need to carry out destructive tests on the green compact. It is more efficient and convenient.

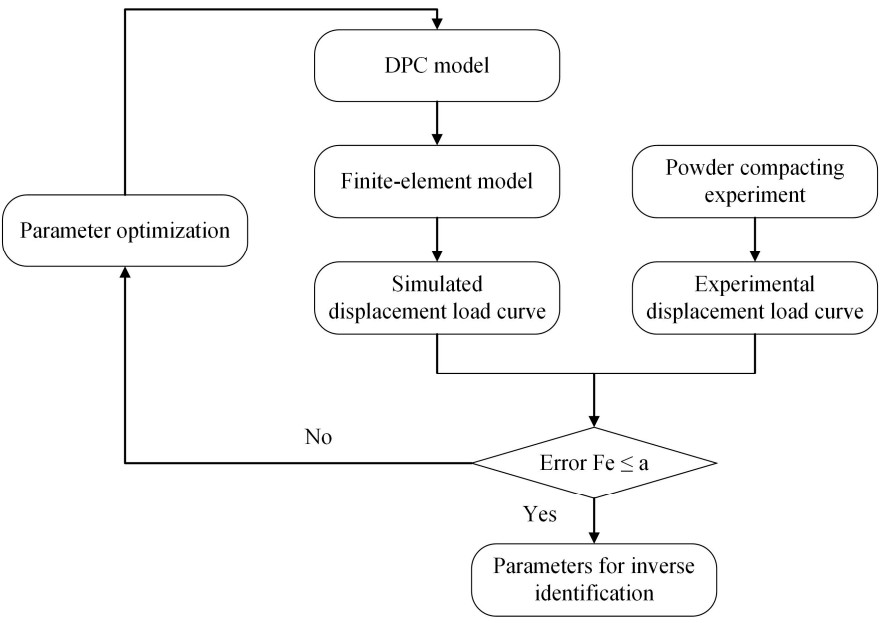

**Figure 6.** Flow chart of the inverse identification of the model parameters.

The parameters of the Drucker–Prager cap constitutive model for the Ti-6Al-4V powder were addressed in this approach for inverse identification. The primary idea is to produce random-beginning-point samples depending on the higher and lower bounds defined for each input variable in the reverse identification process, as well as the characteristics of the optimization algorithm. Then, a simplex optimization algorithm is used to iteratively compute the process, with the minimization of the error function as a recognition objective. During the iterative computing process, the fit of the error function is constantly compared. Individual parameters of the Drucker–Prager cap model are continuously updated to achieve the goal, and the final parameters of the model identified by the inverse identification method are achieved.

It was submitted for computation based on the established flow chart. The inverse identification results for each input variable are shown in Table 3. As shown in Table 3, the inverse identification values of each input variable fall within the specified upper and lower limits. The results do not approach their upper and lower limits, demonstrating the credibility of the inverse identification results. When the results of the inverse identification

are applied to the error function, the error function value is 6.704%. After the calculation of the reverse identification process, the results of the identified variables are better.

**Table 3.** Results of parameter inverse identification.

| Input Variable | Lower Limit | Upper Limit | Reverse Identification Value |
|---|---|---|---|
| d1 | $5.0 \times 10^{-5}$ | $1.0 \times 10^{-4}$ | $7.8992 \times 10^{-5}$ |
| d2 | 12 | 17 | 13.447 |
| m1 | 60 | 70 | 63.232 |
| m2 | 5.0 | 6.0 | 5.4107 |
| m3 | −0.3 | −0.1 | −0.11541 |
| r1 | 0.2 | 0.9 | 0.24013 |
| r2 | 0.9 | 1.3 | 1.1206 |
| b1 | −20 | −10 | −15.628 |
| b2 | $2.0 \times 10^{-2}$ | $6.0 \times 10^{-2}$ | $3.1592 \times 10^{-2}$ |
| b3 | 16 | 20 | 16.689 |
| e1 | −3500 | −2500 | −2926.4 |
| e2 | 3000 | 4000 | 3874.4 |
| e3 | 2 | 4 | 2.2439 |
| v1 | −0.4 | −0.2 | −0.31622 |
| v2 | 0.4 | 0.8 | 0.51028 |
| v3 | 0.2 | 0.5 | 0.37028 |

### 3.2. Numerical Results

The inverse identification parameters were entered into the functional equation in Table 2 to obtain the Drucker–Prager cap model parameters for the Ti-6Al-4V powders, namely, the shear damage surface parameters ($d$ and $\beta$), the cap yield surface parameters ($R$, $P_a$, and $P_b$), and the elasticity parameters (modulus of elasticity $E$ and Poisson's ratio $v$) at various relative densities. Meanwhile, the values of the parameters were entered into the finite-element model for computation, and the displacement–load curve of its finite-element simulation can be obtained. The experimental displacement–load curves were compared to the finite-element modeling displacement–load curves. The comparison findings are displayed in Figure 7. As shown in the figure, the displacement–load curves obtained after parameter inverse identification are in good agreement with the displacement–load curves acquired from the compacting experiment. In general, the trends are consistent. The model parameters obtained by inverse identification can better reflect the mechanical behavior of the Ti-6Al-4V powder during compaction.

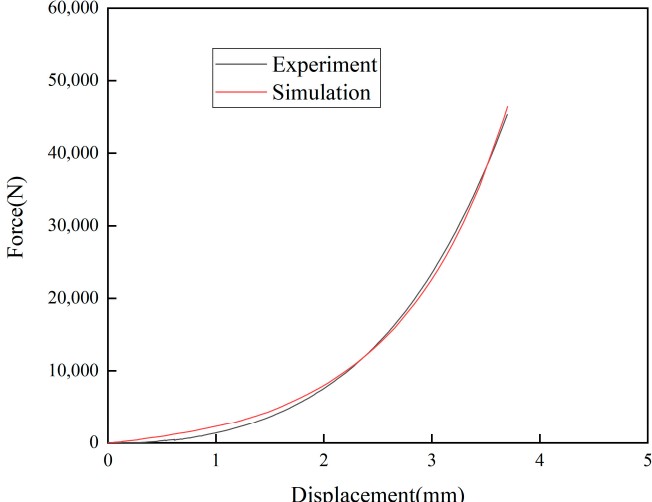

**Figure 7.** Comparison of experimental and simulated displacement–load curves for hemispherical punch compacting.

The inverse identification parameters are input into the finite-element model and numerically simulated. The displacement distribution of the powder after compacting with a hemispherical punch is shown in Figure 8. The "U" represents displacement in millimeters (mm). As can be seen from the figure, the part with a higher displacement is mainly distributed in the area of the powder in contact with the top punch and gradually decreases outward. The relative density distribution of the powder after compacting with a hemispherical punch is shown in Figure 9. From the figure, it can be seen that the relative density distribution of the green compact is not homogeneous. Its higher-density portion is mostly distributed in the area of the powder in contact with the top punch, while the relatively lower-density portion emerges at the edge of the green compact. During the actual compaction process, powder flake occurs at the upper edge of its green compact due to its low relative density. Because the punch has a certain curvature, the center of the powder is first compacted during the compacting process. As the upper punch goes downward, the powder is gradually compacted from the center to the edge, resulting in a reduced relative density at the edge.

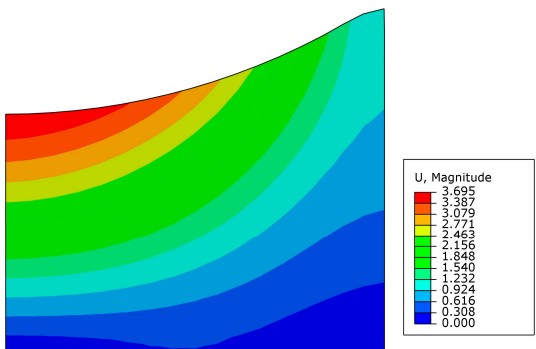

**Figure 8.** Displacement distribution cloud of green compact after hemispherical punch compacting, where the "U" represents displacement in millimeters (mm).

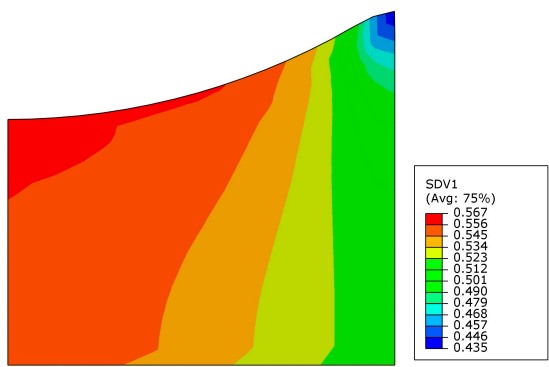

**Figure 9.** Relative density distribution cloud of green compact after hemispherical punch compacting.

Figure 10 illustrates the shear stress, axial stress, and radial stress distributions of the green compact after compacting with a hemispherical punch. The figure shows that the higher-shear-stress portion is mostly localized in the part of the hemispherical punch in contact with the powder and it gradually decreases outward; the maximum place is 35.401 Mpa. Because the upper punch has a specific curvature, the powder in touch with it produced a shear flow under the effect of force, resulting in shear stress. At the same time, the higher-radial-stress portion is mainly concentrated at the center of the green compacting, while the lower portion is concentrated in the upper-right corner of the green compacting. Because the upper punch of the punch has a certain curvature, radial stresses are applied to the powder during compaction. Because of friction and the center of the

powder being compacted first, the radial stress is greatest here. The upper-right corner of the green compacting is the last to make contact with the punch, so the stress is minimized. The powder has a tendency to extend outward during compaction. As a result, the part of the green compact in contact with the cavity die also has a high radial stress. Because the punch goes downward throughout the compacting process, the axial stress in the cloud chart is negative. The figure shows that the axial stress distribution has a gradient. It is primarily shown by higher axial stress in the center of the green compact and lower axial stress towards the edge. In the actual compacting process, the center of the powder contacts with the punch first. The powder is first compacted in the center, so the highest positive stress is found here. As the punch continues to press down, the green compact begins to be gradually compacted from the center area to the edges. As a result, the axial stress gradually decreases. Since the powder is finally compacted at the edges, the axial stress is minimized here. It is proved that the displacement–load curves acquired from compaction tests can be used to identify the shear surface parameters in the constitutive model.

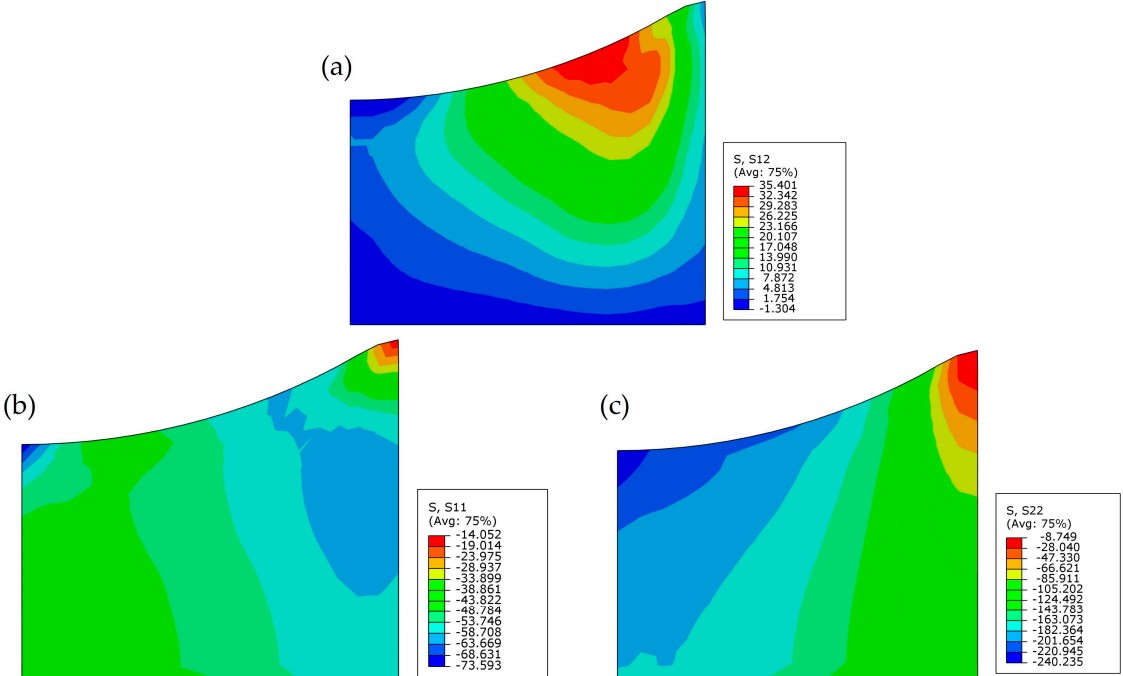

**Figure 10.** The stress distribution cloud of the green compact after hemispherical punch compacting; they should be listed as: (**a**) shear stress; (**b**) radial stress; (**c**) axial stress.

## 4. Experimental Validation by Flat-Punch Compaction

In order to validate the accuracy of the inverse identification parameters, the upper punch was changed to a flat-head shape for the compacting experiments. The displacement–load curve was obtained. Simultaneously, the axisymmetric model was used. It is modeled in the same way as the hemispherical punch. The upper punch, the lower punch, and the cavity die were set up as discrete rigid bodies. The powder was set as a deformable body. At the same time, meshing was performed. The Drucker–Prager cap model parameters obtained through inverse identification were used to set the characteristics of the Ti-6Al-4V powder.

Next, the finite-element simulation was performed. The displacement–load curves obtained from the experiments were compared with the displacement–load curves from the finite-element simulation. The comparison results are shown in Figure 11. From the figure, it can be seen that they have a good agreement. The trend of their two changes is the same, which further validates the accuracy of the inverse identification parameters.

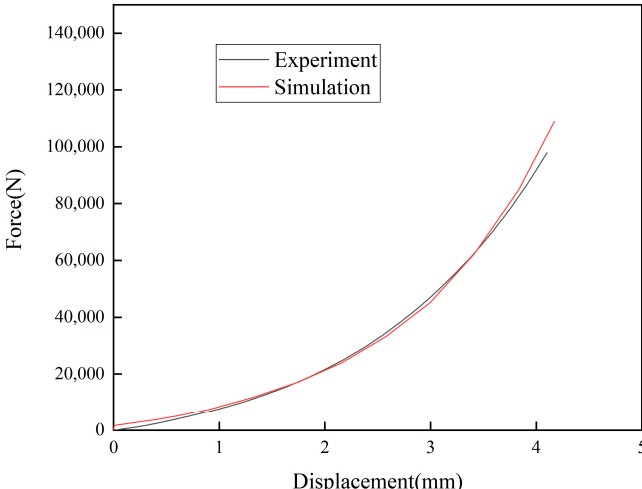

**Figure 11.** Comparison of experimental and simulated displacement–load curves for flat-punch compacting.

The displacement distribution of the green compact after the flat-punch compacting is shown in Figure 12. As can be seen from the figure, the displacement distribution has a gradient. The higher part is mainly located at the top of the powder and decreases gradually toward the bottom. The relative density distribution of the powder after compacting with the flat punch is shown in Figure 13. The green compact of the flat punch has a more uniform relative density distribution. The area of high relative density is found in the upper-right corner, where the upper punch contacts the cavity die. The smaller area occurs in the lower-right corner, where the powder body contacts with the cavity die. It is similar to the results of the actual compacting process. The numerical simulation obtains an average relative density of about 0.590 for the model. The volumetric relative density of the green compact was calculated to be 0.605. It is close to 0.590. There is friction between the powder and the lower surface of the upper punch, the wall of the cavity die, and the surface of the lower punch. Under the combined effect of the pressing force and friction, the powder particles near the contact surface between the green compact and the upper punch move from the center of the cylindrical green compact to the cylindrical surface. The powder particles in the vicinity of the contact surface between the green compact and the lower punch move from the cylindrical surface of the green compact to the center of the green compact. In the compacting direction, the powder particles move in the opposite direction of the movement of the upper punch. Because of the movement of the powder particles, a great number of powder particles congregate in the area where the upper punch contacts the wall of the cavity die. It makes the powder in this area dense. Because the powder particles in the contact area between the lower punch and the cavity die move in the opposite direction of the compaction direction, the relative density of the powder in this region is the lowest.

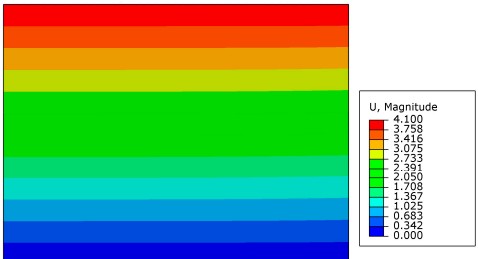

**Figure 12.** Displacement distribution cloud of the green compact after flat-punch compacting.

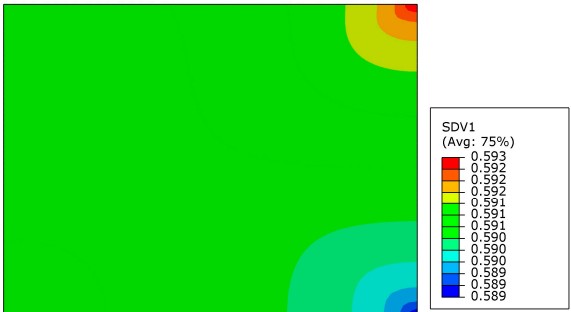

**Figure 13.** Relative density distribution cloud of the green compact after flat-punch compacting.

The distribution of shear stress, axial stress, and radial stress in the green compact of flat punch is shown in Figure 14 below. As shown in the figure, the shear stress of the green compact after flat-punch compacting is mainly concentrated at the interface between the powder body and the cavity die. It decreases towards the center of the powder. Its shear stress is small, at only 2.250 Mpa at the maximum. The distribution of the axial and radial stresses is more uniform. In the actual compacting process, the punch moves downward. Under the action of friction, the powder in contact with the inner wall of the cavity die produces shear flow, resulting in shear stress. However, the shear stress is small. At the same time, the area of the green compact in contact with the upper punch has essentially no shear stress. It indicates that it is not possible to identify the shear surface parameters of the constitutive model using the displacement–load curves obtained from the compaction experiments with the flat punch.

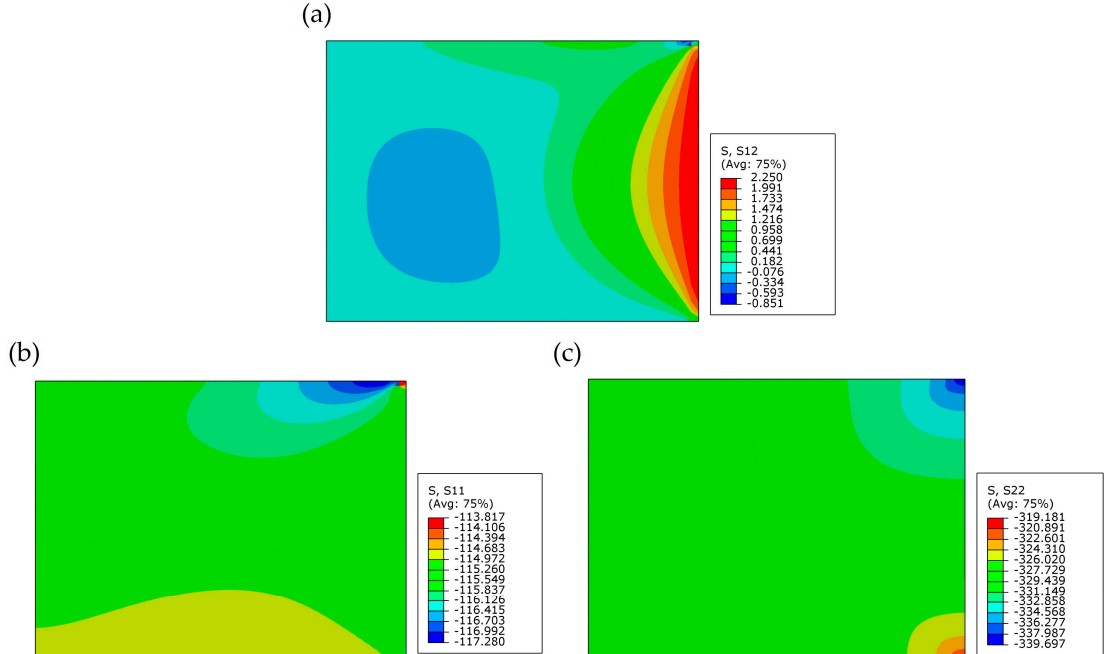

**Figure 14.** The stress distribution cloud of the green compact after flat-punch compacting; they should be listed as: (**a**) shear stress; (**b**) radial stress; (**c**) axial stress.

## 5. Conclusions

(1) This paper proposed an inverse identification approach to quickly and efficiently determine the parameters of the Drucker–Prager cap constitutive model. The approach simply combines the displacement–load curve of the compacting experimental process, establishes the goal error function, and uses the simplex algorithm to compute.

It is more convenient. At the same time, because the traditional parameter inverse identification makes it difficult to identify the shear surface parameters, the shape of the upper punch is creatively changed to a hemispherical punch. The parameters of the constitutive model are identified using the displacement–load curves obtained for powder compaction by a hemispherical punch. It makes the parameters obtained more comprehensive;

(2)  The parameters of the Drucker–Prager cap constitutive model for the Ti-6Al-4V powder material were identified inversely. The displacement–load curves obtained after the inverse identification were compared with the experimentally obtained displacement–load curves. It was found that the two curves were in good agreement. The error function value is 6.704%. At the same time, the relative density distributions were checked. It was found that the relative density distributions obtained from the parameter inverse identification were also more consistent with the actual situation. The feasibility and validity of this inverse identification process were further verified;

(3)  For the advanced powder metallurgy processes, such as hot pressing, hot isotropic pressing, spark plasma sintering, and so on, it is difficult to adopt the conventional semianalytical method for parameter identification of the material model. Therefore, the inverse identification methodology with a hemispherical punch shows great potential for numerical modeling.

**Author Contributions:** Conceptualization, R.L. and W.L.; methodology, R.L.; software, J.L. (Jiaqi Li); validation, J.L. (Jili Liu) and J.L. (Jiaqi Li); formal analysis, R.L.; investigation, R.L.; data curation, R.L.; writing—original draft preparation, R.L.; writing—review and editing, W.L. and J.L. (Jili Liu); visualization, J.L. (Jiaqi Li); supervision, W.L.; project administration, W.L.; funding acquisition, W.L. All authors have read and agreed to the published version of the manuscript.

**Funding:** This research was funded by the National Natural Science Foundation of China, grant number 52005374, and the Open Foundation of the State Key Laboratory of Advanced Technology for Materials Synthesis and Processing (Wuhan University of Technology), grant number 2021-KF-9.

**Data Availability Statement:** Research data are not shared.

**Conflicts of Interest:** The authors declare no conflict of interest.

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
