# Peer review of "Inverse Identification of Drucker–Prager Cap Model for Ti-6Al-4V Powder Compaction Considering the Shear Stress State"

_metals, doi:10.3390/met13111837_

Round 1

Reviewer 1 Report

Comments and Suggestions for Authors

The article presents interesting results on cold compaction of Ti6Al4V powder for identification of the parameters of the Drucker-Prager cap model. I am not a specialist of this field, but it seems to me that the paper is appropriate for publication. My main concern is about language, the article requiring edition by a professional English translator, because some sentences are really difficult to understand. Moreover, as indicated in the remarks below, the meaning of the parameters and their units are difficult to find in the text, or are not indicated. Please, correct this.

Here are some comments.

-          What is a “universal machine”?

-          In table 2, it would be useful to recall the meaning (given in the text) of the different variables: d, beta, R, etc. Same comment for Figure 5: What are d1, d2… and their units?

-          Could you better explain the relations indicated in table 2 (ex: d=d1ed2ρ : is “e” the exponential function?...)

-          For the reader which is not familiar with the formalism of Drucker-Prager, could you recall how its parameters are related to the usual material’s constants: yield strength, elastic modulus, and so on.

-          Figure 5 is difficult to understand without indicating the meaning of the parameters, and there units (for example, is E the Young’s modulus? Therefore, is E = 10000, what are the units?...)

-          Line 238: You probably mean “Table 2” instead of “Table 1”.

-          Fig. 8 : please give the meaning and units of “U”.

-          Due to language issues, the discussions are difficult to follow.

-          Finally, I am surprised (but I am not a specialist) not to find parameters related to the plastic behavior of the powder: yield strength, coefficient of work hardening… Could you comment on that?

Comments on the Quality of English Language

The article requires edition by a professional English translator.

Reviewer 2 Report

Comments and Suggestions for Authors

Overall, the manuscript is well-written and contains some novelty. However, the authors should provide a satisfactory response to the following comments.

The authors should elaborate on findings from previous studies on the use of the Drucker-Prager cap constitutive model

The source of powder should be reported under the materials and method section

Was the mould used for powder compaction purchased commercially or designed for the purpose of this experiement? If the latter applies, what influenced the choice of the design parameters?

Reviewer 3 Report

Comments and Suggestions for Authors

1. What are scientific hypotheses?

2. Are all the equations in the manuscript original or should they be quoted?

3. The authors state which parameters are less and which are more sensitive, but do not explain why this is so.

4. There are two sections listed under number 2 in the manuscript (2. Experiment and simulation of hemispherical punch compaction; 2. Parameter identification of the Drucker-Prager cap model).

5. All input parameters for numerical simulation should be displayed. Their choice should also be justified.

6. How are the lower limit and upper limit values defined in table 3?

7. The parameter inverse identification error function value is 6.704%. Is there a margin of acceptable error?

8. The conclusions should state the innovativeness of the proposed methodology. The disadvantages should also be listed. At the end, future research and possibilities of practical application should be written.

Round 2

Reviewer 3 Report

Comments and Suggestions for Authors

The manuscript has been corrected.

Author Response

We revised the manuscript and you can see the revised version.